# Beyond Binary: Sim-to-Real Dexterous Manipulation with Physics-Aware Contact Representation

*Abstract*—**A primary bottleneck in contact-rich manipulation is the difficulty of collecting real-world data. Sim-to-real reinforcement learning offers a scalable alternative, but the simulation-reality gap prevents information-dense modalities like touch from being effectively used. Existing sim-to-real methods mitigate this gap by simplifying tactile data into low-dimensional features – sacrificing the richness required for complex manipulation. In this work, we introduce CoP, a novel tactile representation grounded in physical principles that preserves dense contact information while maintaining robustness for sim-to-real transfer. To support this representation, we further propose a sensor calibration scheme based on differentiable dynamics, enabling the estimation of taxel orientations without requiring ground-truth force measurements. We evaluate CoP on two challenging contact-rich manipulation tasks: peg-in-hole insertion and ball balancing. Results demonstrate that policies conditioned on CoP achieve successful direct sim-to-real transfer on multi-fingered hands, significantly outperforming both binary and raw tactile baselines. Moreover, the learned policies implicitly capture underlying physical properties, such as object mass, as an emergent byproduct of control. A supplementary video can be found at this URL.**

## I. INTRODUCTION & RELATED WORK

Contact-rich dexterous manipulation remains a central challenge in robotics. While recent advances in teleoperation [1]–[3] and imitation learning from large-scale human demonstrations [4]–[8] have driven impressive progress, these approaches are fundamentally bottlenecked by the high cost and limited scalability of real-world data collection [9]–[11]. Sim-to-real reinforcement learning offers a promising alternative by enabling large-scale training in simulation. However, success has been confined to simple tasks [12]–[16], where the simulation-reality gap can be effectively managed. This gap becomes significantly more pronounced in contact-rich settings, where tactile sensing plays a critical role. Although touch provides information essential for precise and robust manipulation, accurately simulating complex sensor responses that match real-world physical behavior remains notoriously difficult. As a result, high-dimensional tactile signals are often underutilized in sim-to-real pipelines, limiting their applicability to more complex tasks.

At the core of this challenge lies a representation gap. Existing approaches typically fall into two extremes. On one end, simplified tactile features [17]–[19] enable robust sim-to-real transfer but discard rich contact information necessary for complex manipulation. On the other end, learned representations of raw tactile signals [20]–[24] preserve expressiveness but rely heavily on real-world data and lack physical interpretability. This trade-off between representational richness and transfer robustness remains unresolved.

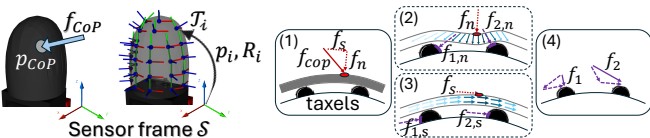

(a) **Left**: CoP contact representation. **Right**: Taxel frames $\mathcal{T}_i$ relative to sensor frame $\mathcal{S}$.

(b) Illustration of the forward mapping based on the proposed stress distribution model to account for surface deformation.

Fig. 1: (a) The CoP representation. (b) The proposed stress distribution model for XELA uSkin [32].

In this work, we address this gap by introducing **CoP**, a physics-grounded tactile representation that preserves dense contact information while remaining robust to sim-to-real discrepancies. Unlike prior approaches, CoP explicitly encodes contact structure in a form that is both expressive and transferable. To support this representation, we further propose a sensor calibration method based on differentiable dynamics, enabling the estimation of taxel orientations without requiring ground-truth force measurements.

Beyond representation, we identify that existing evaluations often obscure the true contribution of tactile sensing, either through heavy reliance on vision [25]–[27] or by focusing on relatively simple, periodic manipulation tasks [28]–[31]. To address this, we introduce two challenging *blind* manipulation tasks—peg-in-hole insertion and ball balancing—that explicitly require tactile feedback while minimizing visual cues. We evaluate our approach through extensive experiments and ablations, demonstrating that policies conditioned on CoP achieve successful direct sim-to-real transfer on multi-fingered hands and significantly outperform both simplified and raw tactile baselines. Furthermore, we show that the learned policies implicitly capture underlying physical properties, such as object mass, highlighting the potential of physically grounded representations for enabling more general and scalable manipulation.

## II. METHODOLOGY

### A. Physics-Aware Contact Representation

We propose **Center-of-Pressure (CoP)** as a physics-aware contact representation. As shown in Fig. 1a (left), CoP consists of a single force vector ${}^{\mathcal{S}}f_{\text{cop}} \in \mathbb{R}^3$ representing the total force acting on the robot link by the object, and the 3-dimensional Cartesian coordinates of a single centroidal contact point ${}^{\mathcal{S}}p_{\text{cop}} \in \mathbb{R}^3$, both expressed in the sensor frame $\mathcal{S}$. While real-world contacts may involve more complex physical interactions, we aim for the CoP representation to be a compact and efficient contact approximation, with the crucial aim of enabling both efficient policy learning in simulation and direct and robust sim-to-real transfer.

## B. Taxel-CoP Mapping

Direct sim-to-real transfer of a learned policy requires the ability to map raw tactile sensor readings on hardware to the CoP representation. Here, we derive a mapping for the XELA uSkin tactile sensors [32], since in this work we use the 16-DOF Allegro hand equipped with the uSkin sensors covering the fingertips, finger phalanges, and palm of the hand. However, we highlight that our proposed CoP representation is applicable to any other sensor with force sensing capabilities. For the XELA uSkin, each sensor array consists of $N$ *taxels* in some grid-like arrangement, where each taxel is an individual tactile sensing point providing 3-axis force measurements. For the $i$-th taxel, the position and orientation of its local coordinate frame $\mathcal{T}_i$ relative to the sensor frame $\mathcal{S}$ are given by ${}^{\mathcal{S}}p_i \in \mathbb{R}^3$ and $R_i \in \mathbb{SO}(3)$ respectively, as shown in Fig. 1a (right). Each taxel force ${}^{\mathcal{T}_i}f_i \in \mathbb{R}^3$ expressed in its local taxel frame $\mathcal{T}_i$ includes both normal and shear components. We denote the inwards, unit surface normal vector at the $i$-th taxel as ${}^{\mathcal{S}}\hat{n}_i \in \mathbb{R}^3$, which can be easily obtained as each taxel is arranged such that one of its frame axes is orthogonal to the surface at its position.

Let ${}^{\mathcal{S}}f_{\text{cop}}$ represent an **unknown** contact force vector on the fingertip, resulting in taxel force activations ${}^{\mathcal{T}_i}f_i$, which are transformed into the sensor frame as ${}^{\mathcal{S}}f_i$. We derive an *invertible*, *differentiable* mapping between raw taxel forces $\{{}^{\mathcal{T}_i}f_i\}_{i=1...N}$ and the CoP representation $\{{}^{\mathcal{S}}f_{\text{cop}}, {}^{\mathcal{S}}p_{\text{cop}}\}$. Here, we focus on the curved fingertip sensor ($N$=30 taxels) since the tasks used in this work only involve fingertip contacts, though the mapping directly generalizes to flat phalanges and palm sensors. From now on, all vector quantities will be defined and expressed in the sensor frame $\mathcal{S}$, therefore we omit frame labeling for better readability.

**Stress Distribution Model**. Due to deformation under external pressure of the thin silicone layer covering the tactile sensors, naively using a weighted sum of the individual taxel forces $f_i$ as the CoP force vector $f_{\text{cop}}$ fails to capture the complex stress distribution to the underlying taxel array. Instead, we propose a simple model as illustrated in Fig. 1b. First, the CoP force vector $f_{\text{cop}}$ is decomposed into its normal $f_n$ and shear $f_s$ components. Then, we model the internal force distribution by accounting for change in force direction due to deformation and force magnitude decay relative to distance from the contact point. For each taxel $i$, this results in *effective* normal $f_{i,n}$ and shear $f_{i,s}$ forces, which are combined into the taxel forces $f_i$.

**Forward Mapping Derivation**. We estimate the CoP position $p_{\text{cop}}$ as the weighted average of individual taxel positions $p_i$ by their measured force magnitudes $\|f_i\|$, where only the set $\mathcal{A} = \{f_i\}$ of *active* taxels with $\|f_i\|$ exceeding a threshold are included. To account for the curved geometry of the fingertip, we approximate the inwards unit surface normal at the CoP, $\hat{n}_{\text{cop}} \in \mathbb{R}^3$, via inverse distance weighting [33] of the taxel normals $\hat{n}_i$, with weights $\alpha_i = \frac{1}{\|p_i - p_{\text{cop}}\|}$.

To model internal stress propagation under deformation, we use a "blended" unit vector $\hat{b}_i$ to approximate the direction of the effective normal force $f_{i,n}$ by interpolating

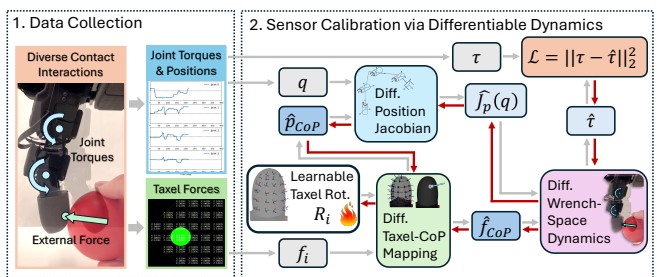

Fig. 2: Our proposed differentiable dynamics-based sensor calibration method, consisting of 1) data collection and 2) gradient-based optimization. Red arrows indicate gradient flow during back-propagation.

between the taxel's local normal $\hat{n}_i$ and the relative unit direction vector $\hat{v}_i = \frac{p_i - p_{\text{cop}}}{\|p_i - p_{\text{cop}}\|}$ from the CoP position to the taxel, $\hat{b}_i = \text{normalize}\,(w_i \hat{n}_i + (1 - w_i)\hat{v}_i)$, where $w_i$ is a Gaussian radial weight $w_i = \exp(-\frac{\|p_i - p_{\text{cop}}\|^2}{2\sigma^2})$, and $\sigma$ is a spread hyperparameter. We approximate the magnitude of $f_{i,n}$ using the same Gaussian decay model on the magnitude of $f_n$. For the effective shear force $f_{i,s}$, we define a shear projection matrix $P_{\text{shear}} = I_3 - \hat{n}_{\text{cop}}\hat{n}_{\text{cop}}^\top \in \mathbb{R}^{3\times3}$ which projects $f_{\text{cop}}$ onto the surface tangent plane at $p_{\text{cop}}$ to approximate the direction of $f_{i,s}$, and use the same Gaussian decay model for the magnitude. Combining everything, the relation between $f_i$ and $f_{\text{cop}}$ can be compactly written as:

$$f_i = M_i f_{\text{cop}}, \quad M_i = w_i(\hat{b}_i \hat{n}_{\text{cop}}^\top + P_{\text{shear}}) \in \mathbb{R}^{3\times3}$$

**Solving for the CoP Force**. To find the unknown $f_{\text{cop}}$ from the observed set of active taxel forces $\{f_i\}$, we aggregate the individual taxel equations into a global linear system $Af_{\text{cop}} = b$, where $A = [M_1^\top, \ldots, M_N^\top]^\top$ and $b = [f_1^\top, \ldots, f_N^\top]^\top$, and obtain $f_{\text{cop}}$ as the closed-form solution to the following regularized least-squares problem:

$$f_{\text{cop}} = (A^\top A + \lambda^2 I)^{-1} A^\top b$$

## C. Sensor Calibration via Differentiable Dynamics

While the taxel origins ${}^{\mathcal{S}}p_i$ are obtainable from the manufacturer's specifications [32], the rotations $R_i$ of the taxel frames $\mathcal{T}_i$ relative to the sensor frame $\mathcal{S}$ are unknown and difficult to manually calibrate due to the complex surface geometry of the fingertip. Here, we present a novel method to learn the taxel orientations via differentiable dynamics.

**Data Collection**. Consider the kinematic chain from the hand's base frame $\mathcal{B}$ to sensor frame $\mathcal{S}$ of a fingertip. To collect data, we first continuously run a stiff PD controller to maintain a fixed set of nominal finger joint positions. Then, random contacts are applied onto the fingertip (see Fig. 2), causing the joint actuators to actively apply torques to balance the external force and maintain static equilibrium. At each time step, the raw taxel forces ${}^{\mathcal{T}_i}f_i$, applied joint torques $\tau \in \mathbb{R}^4$, and joint angles $q \in \mathbb{R}^4$ are recorded.

**Rotation Parameterization**. For choosing how to parameterize the taxel rotations $R_i$ to be learned, we follow recommendations in [34] and use the $\mathbb{R}^9$+SVD method, which parametrizes a rotation by an arbitrary $3\times3$ matrix $P \in \mathbb{R}^{3\times3}$ and projects it to the valid rotation matrix

$R \in \mathbb{SO}(3)$ with the least-squares distance to $P$ using its Singular Value Decomposition (SVD):

$$R = \mathrm{SVD}^+(P) = U\mathrm{diag}(1, 1, \det(UV^\top))V^\top, \ P = U\Sigma V^\top$$

We initialize the learnable rotation parameters as a PyTorch tensor of shape $(N, 3, 3)$ consisting of parameter matrices $P_i \in \mathbb{R}^{3\times3}$ for all $N$ taxels in the sensor array.

**Optimization**. During training, for each data sample, we first rotate the recorded taxel forces $^{\mathcal{T}_i} f_i$ into sensor frame $\mathcal{S}$ using the projected rotations $\hat{R}_i$ based on the current $\hat{P}_i$. Next, the taxel-CoP mapping is applied to obtain the estimated CoP force vector $^\mathcal{S}\hat{f}_{\mathrm{cop}}$ and contact position $^\mathcal{S}\hat{p}_{\mathrm{cop}}$, which are transformed into base frame $\mathcal{B}$ via the forward kinematics computed using the recorded joint angles $q$. Then, we compute the position Jacobian matrix $^\mathcal{B}\hat{J}_{\mathrm{cop}} \in \mathbb{R}^{3\times4}$ at the CoP contact point using $q$ and $^\mathcal{B}\hat{p}_{\mathrm{cop}}$. Under static equilibrium conditions, the following wrench-space relation holds [35], $\tau = -J^\top f + g(q) \approx -J^\top f$, where $f$ is an external force *acting on* the body, $J$ is the position Jacobian at the contact point, and $g(q)$ is a gravity-compensation term that we assume negligible for simplicity. Using this relation, we compute the joint torques $\hat{\tau}$ needed to balance the estimated CoP force $^\mathcal{B}\hat{f}_{\mathrm{cop}}$ as $\hat{\tau} = -^\mathcal{B}\hat{J}_{\mathrm{cop}}^\top {}^\mathcal{B}\hat{f}_{\mathrm{cop}}$. Finally, we compute the MSE loss between the estimated $\hat{\tau}$ and recorded joint torques $\tau$, and backpropagate the gradients to obtain improved estimates of rotation parameters $\hat{P}$.

**Implementation & Training**. We implement the forward pass calculations based on [36] which provides differentiable implementations of forward kinematics and Jacobian matrix calculations. A dataset of 2400 samples (2 minutes at 20Hz control) containing random object-fingertip contacts was collected. We used Adam [37] with a learning rate of 0.1 and performed batch gradient descent for 100 steps. A training visualization is provided in the supplementary video.

Prior works mostly use high-precision force sensors to acquire ground-truth data for sensor calibration [24], [30], [38]–[40]. Instead, our method leverages the differentiability of robot dynamics and our proposed taxel-CoP mapping. Moreover, the framework can in theory learn any differentiable mapping function (e.g. neural network) from raw sensor readings to the CoP representation.

## III. Experiments

**Tasks.** We evaluate on two challenging contact-rich manipulation tasks: **peg-hole insertion** and **ball-balancing**. In both tasks, we train a *blind* policy conditioned only on proprioception (current and commanded joint angles) and contact observations, without any visual input.

**Baselines**. We compare our proposed CoP representation (*cop*) against several baselines, including proprioception (*base*), binary contact per sensing array (*binary*), and CoP force magnitude (*mag*). In addition, we include force vector-only (*vec*) and contact position-only (*pos*) as self-ablated baselines, and raw taxel forces (*taxels*). We also include an expert human (*human*) for comparison.

**Policy Architecture.** A common approach of providing temporal information to the policy is to condition it on

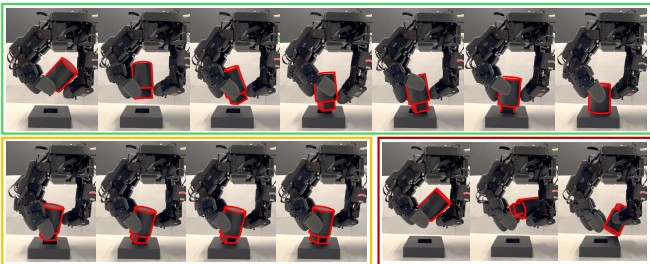

Fig. 3: **Top**: The *cop* policy succeeds under OOD initialization. **Bottom**: The *binary* policy fails to adapt to external contacts (left), and fails to maintain control under OOD initialization (right).

a flattened stack of past observations $o_{t-N:t}$, which has demonstrated effectiveness in robot manipulation [17], [27], [28], [31]. Instead, our policy architecture consists of a single recurrent layer (GRU) which takes as input only the current observation vector $o_t$ and outputs a 256-dim latent vector, which is then passed to a 2-layer MLP with hidden sizes of $[256, 128]$ and ELU activation which outputs the actions. For both tasks, we found that this achieved better sample efficiency and performance over a direct MLP network on stacked history observations (see Appendix A).

**Direct Sim-to-Real Transfer.** We use IsaacLab [41] and train policies using asymmetric actor-critic PPO [42]. The actor observation includes proprioception and the contact representation. The critic additionally receives privileged information such as object states and other task-relevant information. Both networks share the same recurrent architecture. Here, we note that prior works often employ teacher-student distillation [19], [22], [23], [28] due to asymmetric contact representations across simulation and real-world setups, which is more complicated and may lead to degraded performance due to partial observability [43]. In contrast, our aligned CoP representation enables direct sim-to-real transfer (see Appendix B for sim-real alignment details). Full training details are included in Appendix C.

### A. Peg-Hole Insertion

**Task Setup**. We create six pairs of custom peg and hole assets with the same handle and a variety of insertion head shapes (Fig. 11c). The task requires the hand to maintain grasp of the peg and fully insert it into the hole object which is fixed to the table. At each reset, we fully randomize the yaw orientation of the peg, and also introduce small randomizations in its position from manual reset noise. We conduct 10 trials for each observation and each shape, and record the success rate (sr) and task completion time (time).

**Real-World Performance**. As summarized in Table VI, results show that the *cop* policy generally outperforms all baselines in success rate across all insertion shapes. Higher-fidelity contact representations (e.g. *vec*, *cop*) generally lead to more adaptive and persistent policies that achieve higher success rates (Fig. 3, top) but result in longer task completion times than simplified representations (e.g. *base*, *bin*), which achieve fast initial insertions only in certain yaw-initializations while failing to adapt to peg-hole contacts and OOD initializations (Fig. 3, bottom) most likely due to the

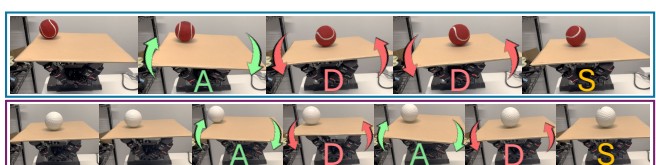
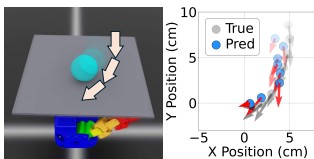

Fig. 4: Two balancing movements from the *cop* policy. "A", "D" and "S" indicate accelerate, decelerate, and stabilize, respectively.

Fig. 5: Predicted and actual ball states in a *cop* policy rollout.

TABLE I: Ball state prediction performance from linear probing of the policy latent embeddings.

|  | x-pos | y-pos | x-vel | y-vel |
|---|---|---|---|---|
| RMSE | 0.013 | 0.019 | 0.059 | 0.065 |
| $r^2$score | 0.76 | 0.62 | 0.23 | 0.15 |

lack of explicit fingertip contact forces. The self-ablations against *vec* and *pos* indicate that both components of the CoP representation are complementary and beneficial towards task performance. Moreover, *taxel* performs worse than nearly all other baselines, which indicates that characteristics in the raw taxel readings may not be fully captured by our tactile simulation. This finding resonates with prior work in tactile simulation for sim-to-real policy transfer, which attempt to alleviate this problem by either training an additional tactile encoder [22] or relying on complex modeling and precise real-world data calibration [24], [44]. Finally, human performance significantly surpasses the best performing robot policy in both metrics, indicating that a significant gap still remains between robot capabilities and human-level dexterity and efficiency in integrating tactile feedback for control.

**Robustness Evaluation**. We repeated the experiment under out-of-distribution peg pose initialization (see Fig. 3). As summarized in Table VI (OOD Init.), while all policies exhibit longer task completion times as more movements are needed to shift the peg downwards towards the hole, the *cop* policy achieves minimal reduction in success rate while exhibiting emergent capabilities of persistent and robust in-hand object translation and re-orientation to achieve peg-hole alignment for successful insertion (Fig. 3, top). Moreover, we repeated the experiment while randomly masking 40% of the raw hardware taxel forces at each time step. As shown in Table VI (Masked), the higher-fidelity contact representations generally suffer from larger performance degradation than the simplified representations, as the former are more sensitive to precise individual taxel forces.

### B. Ball Balancing

**Task Setup**. A lightweight (50g) square plate is supported by the four fingertips, and the hand is required to balance a ball placed on the plate and keep it centered as best as possible. The policy is trained in simulation using a smooth sphere and evaluated in the real world on a variety of four balls with different mass, size, friction, and surface texture (Fig. 11d), resulting in significantly different rolling behaviors outside of the training distribution. We conduct 10 trials for each observation and each ball type, and record the time-to-fall (TTF).

**Real-World Performance**. As shown in Table V, precise force information is crucial for this task, since only *cop*, *vec* and *taxel* policies successfully learned the task (Fig. 11b), with comparable performance between *cop* and *pos* policies, which also suggests that force alone may be sufficient for this task. Fig. 4 shows two distinct emergent movement patterns, including a more aggressive single-step accelerate-decelerate maneuver (top) and a slower two-step

centering process (bottom). Interestingly, while the robot policy struggles more with the smoother and faster-rolling hockey ball, the *human* struggles the most with the dented moon ball which exhibits nonlinear, unpredictable rolling. As suggested by prior research, the human may be extrapolating the current observations for future prediction using a *linear* model [45], [46]. In contrast, the robot policy may be more reactive towards the immediate state rather than extrapolating future information for control.

**Object State Prediction**. To understand how a trained policy leverages the explicit CoP contact information for control, we analyze the 256-dim latent output of the recurrent layer of the policy network. We first collected a training set of 1000 5s trajectories (100 samples each), and performed *linear probing* to predict the ball state consisting of position $p \in \mathbb{R}^2$ and velocity $v \in \mathbb{R}^2$ in the xy-plane. Then, we collected a test set of 100 5s trajectories, and compute the RMSE and $r^2$ scores between the true states and model predictions. As shown in Table I, the policy's latent representation effectively captures the ball's position. However, velocity prediction is noticeably weaker, which indicates that while the policy leverages contact information to track ball position, it may not encode motion dynamics as precisely, likely due to the inherent noise in contact-based state estimation. Fig. 5 illustrates the predicted and actual ball states in an example policy rollout trajectory.

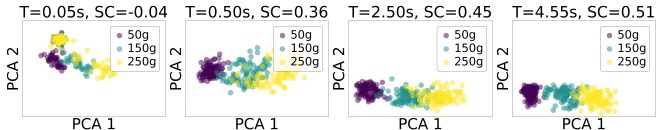

Fig. 6: Visualization of the emergence of mass-correlated latent embedding clusters across *cop* policy rollout trajectories temporally. SC indicates the Silhouette Coefficient.

**Implicit Mass Identification**. We further examine whether a trained policy's latent representation implicitly captures dynamical properties of the ball such as its mass. We choose 3 different ball masses (50g, 150g, 250g) and collect 100 5s trajectories for each mass. Then, we perform PCA to transform all 256-dim latent embeddings into their first two principle components. Fig. 6 reveals that as trajectories evolve over time, the latent embeddings naturally reorganize into distinct clusters corresponding to different mass values. The gradual increase in the Silhouette Coefficient (SC) further suggests that the policy's recurrent layer progressively disambiguates physical constants such as mass. It also demonstrates that conditioning on the CoP contact representation allows the network to discover underlying object properties as a byproduct of solving the control task, without explicit or teacher-student supervision.

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

### A. Policy Network Architecture Comparison

We analyze the performance of our recurrent policy architecture consisting of a single recurrent layer (GRU) which outputs a 256-dim latent that is passed to two linear layers with sizes $[256, 128]$ and ELU activation, and compare it against a direct MLP with three linear layers with sizes $[512, 256, 128]$ and ELU activation. Moreover, to examine whether providing longer temporal context via stacking more history observations to a MLP policy improves performance, we train several policies conditioned on different numbers $(3, 5, 10, 20)$ of history steps of concatenated observation vectors. The PPO hyperparameters and training configurations are kept identical for the recurrent and MLP policies to ensure fairness (see Appendix C.5). As shown in Fig. 7, the performance gains saturate for MLP policies conditioned on longer observation histories. For both tasks, we found that the recurrent policy results in better sample efficiency and convergence quality than the MLP policy. One possible underlying factor for this result is that for incorporating longer-horizon temporal information, the recurrent layer maintains the same observation space dimension, whereas the direct MLP network requires concatenating larger numbers of past observation vectors and thus drastically increasing the observation space dimension. Another possible factor may be that the recurrent layer's update mechanism fuses the current observations with its internal hidden state, which may act as a "regularizer" and effectively smooths out noisy observations.

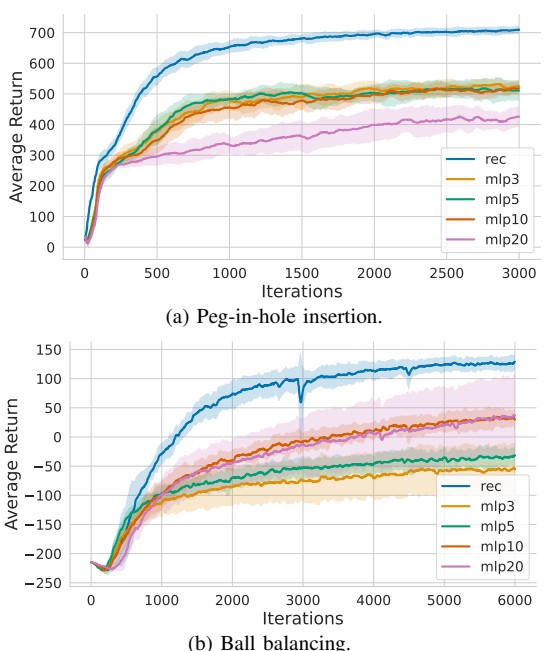

(a) Peg-in-hole insertion.

(b) Ball balancing.

Fig. 7: Comparison of training performance between using a recurrent policy network and MLP networks with different history lengths of stacked input observation vectors as input. Shaded area represents the standard deviation over 5 random seeds.

### B. Sim-Real Alignment

*1) Contact Observation Alignment:* We use the ContactSensor API in IsaacLab [41] to track contacts

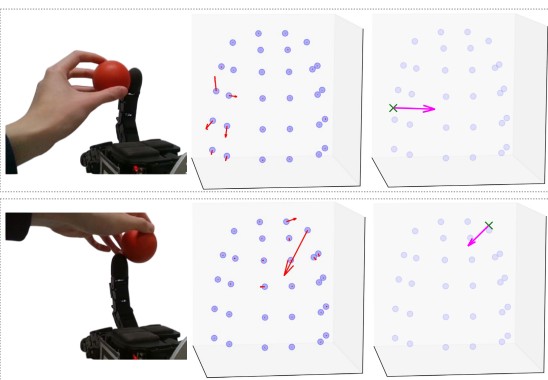

Fig. 8: Visualizations of the computed CoP contact (green cross & pink arrow) from raw taxel forces (red arrows). In each example, the middle subplot shows the raw taxel force vectors, and the right subplot shows the computed CoP force vector and contact position.

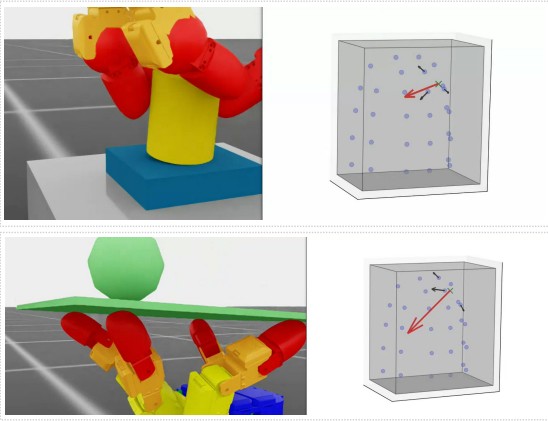

Fig. 9: Visualizations of the computed taxel force vectors (black arrows) from CoP contacts (green cross & red arrow) given by the simulator. Grey boxes represent the taxel-covered regions which are used to filter contacts during simulation.

between the taxel-covered regions of each fingertip link and the contact object. Preliminary results revealed that the simulated contact forces are predominantly normal to the fingertip surface with minimal shear components. Therefore, we restrict $f_{\mathrm{cop}}$ in simulation to its surface-normal component, and on hardware we project the computed $f_{\mathrm{cop}}$ onto the inwards normal $\hat{n}_{\mathrm{cop}}$ as an additional step in the taxel-CoP mapping. To achieve approximate alignment between simulation and hardware, we recorded tactile and contact observations from policy rollout trajectories and carefully tuned the hyperparameters of the taxel-CoP mapping, which we found to work surprisingly well.

The proposed invertible taxel-CoP mapping enables us to both convert raw taxel activations into the CoP contact representation and compute taxel forces from a CoP contact interaction. Fig. 8 visualizes the computed CoP force vector and contact position from raw taxel forces recorded on hardware during real-world contact interactions with an external object. Fig. 9 visualizes the CoP contact information provided by the simulator and the computed raw taxel force vectors using the taxel-CoP mapping, where both show the forces on the middle fingertip as an example.

*2) System Identification of Actuator Dynamics:* Prior work have achieved enhanced sim-to-real transfer perfor-

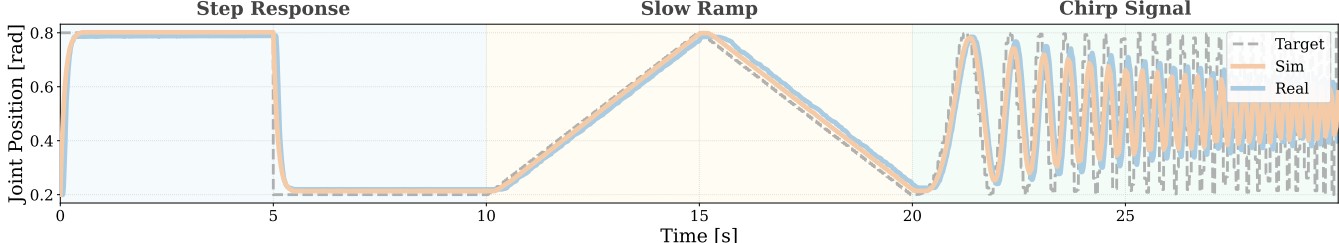

Fig. 10: Aligned actuator dynamics across simulation and hardware.

mance through real-to-sim system identification of actuator dynamics [16], [47]. Inspired by this, we design a Bayesian optimization-based approach to identify the combination of configurable actuator parameters in simulation that results in the best alignment with hardware actuator dynamics. In IsaacSim, the key configurable actuators parameters include stiffness, damping, and joint friction. To find the set of simulation parameters that results in actuation behaviors closely-aligned to the real hardware, we designed a series of trajectory sequences to probe the dynamic responses of each actuator, including step inputs, slow linearly-varying ramp inputs, and chirp inputs with increasing frequency, as shown in Fig. 10. Then, we choose a fixed set of actuator parameters, apply onto the real robot, and execute the trajectory sequences and record the trajectory data consisting of joint angles at all time steps. Next, we begin an automatic tuning process in simulation via a Bayesian optimization loop. At each iteration, the current sampled joint parameters are applied to the simulated robot, the same trajectory sequences are executed, and the trajectory data is recorded. We then compute the MSE loss between the sim and real trajectories and use it as the minimization objective for the Bayesian optimizer to sample the parameters in the next iteration. Fig. 10 shows an example of trajectory alignment after optimizing for the actuator parameters of a given joint.

*3) Sensor Delay:* The tactile sensors introduce non-negligible delay from time-of-contact to the policy receiving it as observation, which is critical for highly dynamic tasks. We measure sensor delay via vision-based methods and introduce it into simulation during policy training.

### C. RL Configuration & Training Details

*1) Actor & Critic Observations:* We employ PPO with asymmetric actor and critic observations for policy training. At each time step, the policy observation includes the current joint angles $q_t$, previous actions $a_{t-1}$, and the contact representation. For contact representations such as *cop* and *taxel* which are stored as higher-order tensors, we flatten them into a single vector and concatenate it to the proprioception vector. The critic observation includes privileged task-relevant information in addition to the actor observations, and are summarized in Table II.

*2) Policy Action Space:* At each time step, the policy network outputs 16-dim actions $a \in \mathbb{R}^{16}$, which correspond to target joint position *increments* across all 16 DOFs. We first clip the actions to the range $[-1, 1]^{16}$, and apply an action scale of 0.03 for peg-hole insertion and 0.05 for ball

TABLE II: Summary of privileged critic observations for both tasks.

| Peg-Hole Insertion | | Ball Balancing | |
|---|---|---|---|
| observation | quantity | observation | quantity |
| peg position | $p_{\text{peg}} \in \mathbb{R}^3$ | plate position | $p_{\text{plate}} \in \mathbb{R}^3$ |
| peg rotation | $R_{\text{peg}} \in \mathbb{R}^4$ | plate rotation | $R_{\text{plate}} \in \mathbb{R}^4$ |
| goal vector | $r_{\text{goal}} =$ $p_{\text{goal}} - p_{\text{peg}} \in \mathbb{R}^3$ | plate linear velocity | $v_{\text{plate}} \in \mathbb{R}^3$ |
| goal reached | $\mathbb{1}(\|r_{\text{goal}}\| \leqslant \epsilon)$ | plate angular velocity | $\omega_{\text{plate}} \in \mathbb{R}^3$ |
| | | ball position | $p_{\text{ball}} \in \mathbb{R}^3$ |
| | | ball linear velocity | $v_{\text{ball}} \in \mathbb{R}^3$ |
| | | goal distance | $d_{\text{goal}} =$ $p_{\text{ball}} - p_{\text{plate}} \in \mathbb{R}^3$ |

TABLE III: Summary of reward terms for both tasks.

| Reward Term | Expression | Weight |
|---|---|---|
| **Peg-Hole Insertion** | | |
| goal distance reward | $\exp(-0.5(d_{\text{goal}}/0.015))$ | 1.0 |
| goal reached reward (given once) | $\mathbb{1}(\|r_{\text{goal}}\| \leqslant \epsilon)$ | 400.0 |
| good contact reward | $\sum_{i=1}^{N_{\text{sensors}}} \mathbb{1}(\|f_i\| \geqslant 1.0)$ | 0.25 |
| rotation deviation (from z-axis) penalty | $1 - \exp(-0.5(\|R_{\text{peg},z}\|/0.6)^2)$ | 1.0 |
| hand DOF pos deviation penalty | $1 - \exp(-0.5(\|q - q_{\text{default}}\|/0.2)^2)$ | 1.0 |
| **Ball Balancing** | | |
| goal distance reward | $\exp(-d_{\text{goal}}/0.05)$ | 1.0 |
| plate contact reward | $\sum_{i=1}^{N_{\text{sensors}}} \mathbb{1}(\|f_i\| \geqslant 0)$ | 0.2 |
| ball-plate relative linear velocity penalty | $\tanh(\|v_{\text{ball}} - v_{\text{plate}}\|/0.3)$ | 1.0 |
| plate position deviation penalty | $\tanh(\|p_{\text{plate}} - p_{\text{plate,default}}\|/0.15)$ | 1.0 |
| plate yaw penalty | $\tanh(\|R_{\text{plate},z}\|/0.5)$ | 1.0 |
| ball fallen penalty | $\mathbb{1}(p_{\text{ball},z} \leqslant 0.2)$ | 200.0 |
| action diff penalty | $\tanh(\|a_t - a_{t-1}\|^2/16.0)$ | 1.0 |

balancing. Then, we apply an exponential moving average (EMA) with $\alpha = 0.5$ to the scaled target joint position increments, and add it to the previously commanded joint target positions to obtain the current joint targets which are then tracked by the PD controller. For the PD controller gains, we use $P = 3.0, D = 0.1$ for the insertion task for more compliant control, and $P = 6.0, D = 0.15$ for the balancing task for more reactive control.

*3) Rewards:* The reward terms for each task consists of task progress-based reward terms and regularizing penalty terms, and are summarized in Table III.

*4) Domain Randomization:* We employ systematic domain randomization to object physical properties including friction and mass, object states during resets, and observation noise and delay to make the policy robust against noisy real-world variations. Domain randomization setups for both tasks are summarized in Table IV.

*5) PPO Configuration & Training Results:* We use the Proximal Policy Optimization (PPO) algorithm to learn RL policies. To ensure stable learning across varying feature scales, we apply online observation normalization to both the actor and critic inputs. The training uses an adaptive learning rate schedule starting at $5.0 \times 10^{-4}$ with a target KL divergence of 0.016. During each iteration, we collect

TABLE IV: Domain randomization setup including both task-specific and general configurations.

| Peg-Hole Insertion | |
| --- | --- |
| Peg: Mass (kg) | $[0.03, 0.04]$ |
| Peg: Friction (static & dynamic) | $[0.2, 0.4]; [0.1, 0.2]$ |
| Peg: Initial Roll & Pitch (rad) | $+\mathcal{U}(-0.2, 0.2)$ |
| Peg: Initial Yaw (rad) | $+\mathcal{U}(0, 2\pi)$ |
| Hole: Friction (static & dynamic) | $[0.3, 0.5]; [0.1, 0.3]$ |
| Hand: Friction (static & dynamic) | $[0.5, 0.7]; [0.3, 0.5]$ |
| Hand: Initial Position (cm) | $+\mathcal{U}(-0.5, 0.5)$ |
| Hand: Initial Roll & Pitch (rad) | $+\mathcal{U}(-0.02, 0.02)$ |
| Hand: Initial DOF pos (rad) | $+\mathcal{U}(-0.05, 0.05)$ |
| Ball Balancing | |
| Ball: Mass (kg) | $[0.05, 0.25]$ |
| Ball: Friction (static & dynamic) | $[0.01, 0.02]; [0.0, 0.01]$ |
| Ball: Initial Position (cm) | $+\mathcal{U}(-5.0, 5.0)$ |
| Plate: Mass (kg) | $[0.035, 0.055]$ |
| Plate: Friction (static & dynamic) | $[0.01, 0.02]; [0.0, 0.01]$ |
| Plate: Initial Position (cm) | $+\mathcal{U}(-0.5, 0.5)$ |
| Hand: Friction (static & dynamic) | $[1.9, 2.0]; [1.8, 1.9]$ |
| Hand: Initial Roll & Pitch (rad) | $+\mathcal{U}(-0.02, 0.02)$ |
| Hand: Initial DOF pos (rad) | $+\mathcal{U}(-0.05, 0.05)$ |
| General | |
| PD Controller: P Gain | $\times\mathcal{U}(0.8, 1.2)$ |
| PD Controller: D Gain | $\times\mathcal{U}(0.7, 1.3)$ |
| Joint Pos Obs Noise (rad) | $+\mathcal{U}(-0.1, 0.1)$ |
| Force Vector Obs Noise: Prob | $0.2$ |
| Force Vector Obs Noise: Rotation (rad) | $+\mathcal{U}(-0.1, 0.1)$ |
| Force Vector Obs Noise: Magnitude | $\times\mathcal{U}(0.9, 1.1)$ |
| Contact Pos Obs Noise Prob | $0.2$ |
| Contact Pos Obs Noise (cm) | $+\mathcal{U}(-0.1, 0.1)$ |
| Joint Pos Obs Delay (s) | $0.05$ |
| Contact Obs Delay (s) | $[0.05, 0.1]$ |

64 steps for the insertion task and 16 steps for the balancing task per environment and perform 5 epochs of updates over 4 mini-batches. We use discount factor $\gamma = 0.99$, GAE parameter $\lambda = 0.95$, clipping range of 0.2, and entropy coefficient of 0.005 to encourage exploration.

Fig. 11 shows the training results for both tasks. In the peg-hole insertion task, the *cop* policy on average attains the highest rewards during training than all other observations, though the difference is fairly marginal. For this task, the high exploration noise during training may have led to successful insertions through action stochasticity, thus artificially inflating the rewards of less robust policies. In contrast, as shown in the real-world transfer results, during inference, the cop observation enables the policy to achieve more consistent and robust performance than other observations. For the ball balancing task, the policies conditioned on contact representation which lack explicit force information, including *base*, *bin*, and *pos*, completely fails to learn the task. Consistent with the real-world results, we observe that the *taxel* observation results in worse performance than more compact contact representations such as *force* and *cop*, which is likely due to the high dimensionality and complexity of the raw taxel forces.

### D. Real-World Performance

The zero-shot sim-to-real transfer performance for the ball balancing and peg-in-hole insertion tasks are detailed in Tables V and VI, respectively.

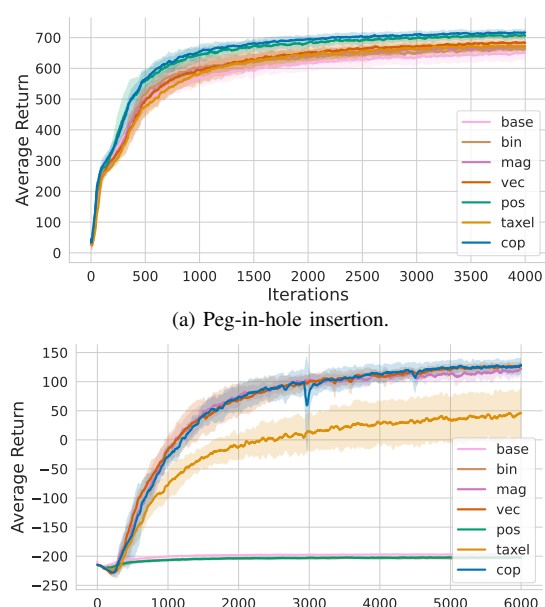

(a) Peg-in-hole insertion.

(b) Ball balancing.

Fig. 11: Comparison of training performance across different contact representations for policy observation. Shaded area represents the standard deviation over 5 random seeds.

TABLE V: Performance comparison of different contact representations for the ball balancing task. We report time-to-fall (s) across 10 trials for each condition and each ball type.

| | Tennis ball↑ | Baseball↑ | Moon ball↑ | Hockey ball↑ | Overall↑ |
| --- | --- | --- | --- | --- | --- |
| base | $1.38_{\pm0.17}$ | $1.42_{\pm0.15}$ | $1.50_{\pm0.22}$ | $1.24_{\pm0.16}$ | $1.38_{\pm0.21}$ |
| bin | $2.20_{\pm1.02}$ | $2.22_{\pm1.44}$ | $1.78_{\pm0.71}$ | $1.75_{\pm0.45}$ | $1.99_{\pm1.03}$ |
| mag | $2.83_{\pm0.97}$ | $2.17_{\pm0.48}$ | $2.35_{\pm0.36}$ | $2.25_{\pm0.90}$ | $2.40_{\pm0.79}$ |
| vec | $5.59_{\pm3.52}$ | $3.27_{\pm0.41}$ | $4.59_{\pm1.11}$ | $2.80_{\pm0.91}$ | $4.52_{\pm2.93}$ |
| pos | $1.63_{\pm0.15}$ | $1.59_{\pm0.30}$ | $1.70_{\pm0.16}$ | $1.26_{\pm0.14}$ | $1.55_{\pm0.27}$ |
| taxel | $1.38_{\pm0.33}$ | $1.73_{\pm0.31}$ | $1.61_{\pm0.34}$ | $1.22_{\pm0.10}$ | $1.49_{\pm0.36}$ |
| cop (ours) | $\mathbf{5.07}_{\pm2.13}$ | $\mathbf{4.77}_{\pm1.55}$ | $\mathbf{4.50}_{\pm1.29}$ | $\mathbf{3.06}_{\pm0.64}$ | $\mathbf{4.60}_{\pm2.19}$ |
| human | $11.29_{\pm6.55}$ | $9.41_{\pm4.46}$ | $5.96_{\pm0.78}$ | $10.82_{\pm5.18}$ | $9.37_{\pm5.32}$ |

### E. Task Asset Details

For the insertion task, we generated custom peg and hole assets using six different shapes – circle, diamond, ellipse, hexagon, square, and triangle – as shown in Fig. 11c. Fig. 13 details their dimensions. For the corresponding hole assets, the sizes of the hole are increased by 10% in the $x$ and $y$ axes, resulting in 10% error tolerance for insertion. This was motivated by the observation that in simulation, using zero tolerance induces "jamming" behaviors between the head of the insertion peg and the hole during the insertion process, likely due to mesh approximations and potential numerical errors in the physics engine's collision resolution.

For the ball balancing task, we evaluate using a light cardboard (mass = 45g) and four different balls with varying size, mass, surface geometry and friction, as shown in Fig. 11d. During policy training in simulation, we use a fixed-size ball with a radius of 6cm, and randomize its mass within a chosen range that captures the mass distribution of the real objects.

TABLE VI: Performance comparison of different contact representations for the insertion task. We report success rate (%) and time-to-fall (s) across 10 trials for each condition and each insertion shape.

| | Circle sr↑ | Circle time↓ | Diamond sr↑ | Diamond time↓ | Ellipse sr↑ | Ellipse time↓ | Hexagon sr↑ | Hexagon time↓ | Square sr↑ | Square time↓ | Triangle sr↑ | Triangle time↓ | Overall sr↑ | Overall time↓ | OOD Init. sr↑ | OOD Init. time↓ | Masked sr↑ | Masked time↓ |
|---|---|---|---|---|---|---|---|---|---|---|---|---|---|---|---|---|---|---|
| base | 0.8 | 5.36 ±0.79 | 0.2 | 4.28 ±0.35 | 0.3 | 2.85 ±0.11 | 0.6 | 2.54 ±0.28 | 0.4 | 2.72 ±0.74 | 0.3 | 10.36 ±1.31 | 0.43 | 4.65 ±2.80 | 0.17 | 11.76 ±6.60 | - | - |
| bin | 1.0 | 2.60 ±0.34 | 0.2 | 10.82 ±3.54 | 0.3 | 2.58 ±0.87 | 0.8 | 8.30 ±3.42 | 0.6 | 16.73 ±4.03 | 0.3 | 26.29 ±0.32 | 0.53 | 10.15 ±8.57 | 0.20 | 14.40 ±5.83 | 0.52 | 14.87 ±17.30 |
| mag | 1.0 | 3.10 ±0.34 | 0.5 | 15.94 ±7.88 | 0.4 | 3.95 ±1.25 | 0.8 | 13.03 ±7.90 | 0.5 | 14.08 ±10.53 | 0.1 | 23.50 ±0.00 | 0.55 | 9.47 ±9.73 | 0.27 | 19.33 ±7.00 | 0.48 | 11.13 ±16.12 |
| vec | 1.0 | 2.97 ±0.64 | 0.4 | 9.32 ±6.54 | 0.5 | 6.89 ±1.30 | 1.0 | 8.06 ±10.29 | 0.7 | 3.43 ±1.98 | 0.4 | 19.56 ±2.60 | 0.67 | 7.19 ±7.60 | 0.42 | 20.79 ±5.07 | 0.57 | 13.09 ±14.58 |
| pos | 1.0 | 2.56 ±0.30 | 0.3 | 15.59 ±10.96 | 0.2 | 27.62 ±6.31 | 0.5 | 4.71 ±0.31 | 0.6 | 7.68 ±4.28 | 0.4 | 24.40 ±1.00 | 0.50 | 10.19 ±10.12 | 0.28 | 11.96 ±8.01 | 0.48 | 8.12 ±10.53 |
| taxel | 0.8 | 9.02 ±9.78 | 0.2 | 20.89 ±8.26 | 0.4 | 5.57 ±0.17 | 0.6 | 2.94 ±0.56 | 0.6 | 17.65 ±6.34 | 0.3 | 18.08 ±10.36 | 0.48 | 10.94 ±9.81 | 0.27 | 15.44 ±10.07 | 0.30 | 9.09 ±14.33 |
| cop (ours) | **1.0** | **4.22** ±0.97 | **0.6** | **14.89** ±7.91 | **0.6** | **7.61** ±4.85 | **1.0** | **6.06** ±2.99 | **0.9** | **13.61** ±6.11 | **0.6** | **20.37** ±6.39 | **0.78** | **10.34** ±7.62 | **0.63** | **15.21** ±6.54 | **0.62** | **12.70** ±14.71 |
| human | 1.0 | 0.78 ±0.19 | 1.0 | 2.99 ±1.13 | 1.0 | 2.49 ±1.32 | 1.0 | 1.17 ±0.30 | 1.0 | 1.69 ±0.94 | 1.0 | 3.07 ±1.19 | 1.0 | 2.03 ±1.32 | - | - | - | - |

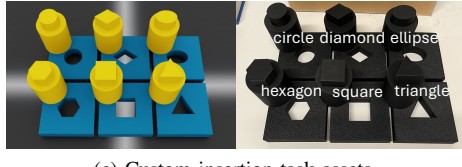

(c) Custom insertion task assets.

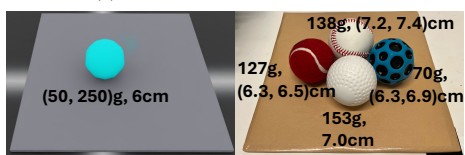

(d) Ball balancing task assets.

Fig. 12: Simulation and real-world assets for the insertion (a) and ball-balancing (b) tasks.

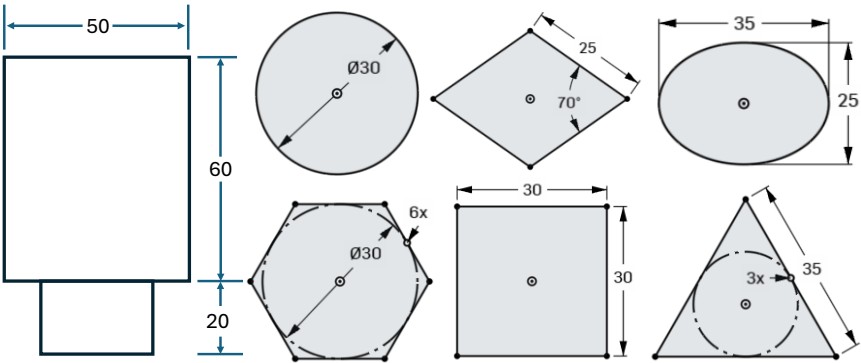

Fig. 13: **Left**: Dimensions of the insertion peg. **Right**: Dimensions of each insertion shape (mm).