# OpenReview forum: "Beyond Binary: Sim-to-Real Dexterous Manipulation with Physics-Aware Contact Representation"
_IEEE.org/ICRA/2026/Workshop/Manipulation_Robustness — ICRA 2026_

### Official Review · Reviewer_JvoS · 2026-05-13
**Review of "Beyond Binary: Sim-to-Real Dexterous Manipulation with Physics-Aware Contact Representation"**

**Rating:** 8
**Confidence:** 5

**Review:**

### **Summary**
This paper proposes CoP, a simple physics-based tactile representation that maps raw taxel forces into a single contact force and contact point for sim-to-real transfer. The method is evaluated on two blind manipulation tasks: peg-in-hole insertion and ball balancing, where CoP-conditioned policies outperform both simplified and raw tactile baselines.
### **Pros**

- **Comprehensive and rigorous experimental setup.** The paper includes a complete set of baselines spanning the full information spectrum.
- **Learning-free, physics-grounded tactile representation.** Unlike prior learned encoders, CoP uses an analytical, invertible, differentiable mapping that requires no training data and yields physically interpretable outputs. This design naturally aligns simulation and real, enabling direct sim2real transfer without teacher-student distillation.
- **Task design effectively proves tactile contributions.** By choosing blind manipulation tasks and removing visual input entirely, the paper exposes the true contribution of different tactile representations. The ball balancing task is particularly discriminative: methods lacking explicit force information completely fail to learn.
- **Compelling emergent phenomena.** The implicit mass identification result is striking: the policy spontaneously organizes its latent space by ball mass despite never receiving mass as input. This suggests CoP-conditioned policies perform implicit system identification as a byproduct of control, potentially reducing the need for teacher-student distillation.

### **Cons**

- **Multi-point contact is not addressed.** The CoP representation is fundamentally a low-order approximation of the contact force distribution and fails when multiple separated contacts occur within a single fingertip. The paper sidesteps this through task choices (per-fingertip single-point contact) and likely single-point calibration data, but the limitation is not openly discussed.
- **Position component in COP appears unnecessary in the chosen tasks.** Self-ablations show “vec” matches “cop” on ball balancing and nearly matches it on peg-insertion, while “pos” alone barely beats the proprioception baseline. A task that genuinely requires fine-grained contact localization would be needed to justify keeping the position component.
- **Calibration precision is never directly evaluated.** The differentiable dynamics calibration is presented as a key contribution, but is validated only through downstream task performance. An ablation against uncalibrated baseline is needed to show whether this step actually matters.
- **Do we really need history information?** Appendix A compares GRU to stacked MLP variants but omits a single step MLP baseline, leaving open whether the recurrent policy's benefit comes from history integration or from architectural inductive biases. A comparison against a memoryless baseline would clarify this.
- **No qualitative failure mode analysis.** The paper reports aggregate metrics but does not characterize what failures look like. Failure videos or text descriptions would clarify the practical boundaries of the method.

---

### Decision · Program_Chairs · 2026-05-21

Accept